# Ascending: An Exploration of Women's Leadership Advancement in the Role of Board of Trustee Chair

Heather Inez Ricks Scott

Department of Leadership and Integrative Studies, Kennesaw State University, Kennesaw, GA 30144, USA; hscott18@kennesaw.edu

**Abstract:** While women have made strides in leadership in the higher education sector there continues to be dismal representation of women in executive level roles of governance at colleges and universities. This article presents findings from a study that explored skills that women have identified as being useful in their ascent to the role of trustee board chair. The ascension patterns of the participants are explored through a qualitative process to provide a path to success for other women to follow. The article concludes with suggestions for increasing the number of women serving in the capacity of board chair.

**Keywords:** women and higher education; leadership; governing boards

## 1. Introduction

The trustee board plays a critical role in the stability, growth, and financial strength of colleges and universities (Baldridge et al. 1977). The board of trustees, along with the president, set the tone for the vision of an institution. The board chair plays a vital part in developing the culture of the board. The knowledge and experience these individuals contribute to the institution have a tremendous impact on the leadership effectiveness of the institution. Board positions are powerful policy making positions and the individuals who hold the chair role are extremely influential. In addition to the policy influence associated with the board chair position, the board chair position is also instrumental in creating a representation of the institution's governance at its highest level (Bowen 2008).

This work presents information from a study on collegiate governing board chairs. The study was conducted to determine the career paths taken, obstacles overcome along those paths, and assistance received along the way in an effort to develop a paradigm of ascension patterns for women trustee board chairpersons. Exploring the path of women to leadership service in the role of board chair adds to the body of knowledge regarding leadership and gender at the policy-making level of collegiate institutions. A portion of the findings from a larger study are presented here with particular focus on pathways to the role of board chair, and specific attention to the participant's own assessments of how their contributions as board members led to their accession to their chair positions.

With continued low numbers of women represented in leadership roles on the governing boards of both public and private higher learning institutions in the United States, it is important to identify the contributing factors that impact women's ascension to the role of board chair (Bontrager 2008; Glazer-Raymo 1999; Kaufman 2002; Schwartz and Akins 2004). Research assists in providing more insight into the role of women as leaders in higher education and the pathways that women take to their respective leadership roles.

The role of women in higher education has seen numerous evolutions throughout the history of academe. Much progress has been made in the education of women since Colonial times when it was considered unfeminine and anti-family for females to seek academic pursuits (Wood 2009). While American higher education dates back to the founding of Harvard College (1636),

women were not allowed college entry until the 19th Century when Oberlin College allowed women to be admitted (Wood 2009). Despite National Center for Education Statistics data that shows women continually outpacing men in degree attainment, there is still not parallel representation of women in collegiate leadership roles in comparison to their male peers. The 2016 Digest of Education of Statistics reports that in 2014–2015 women earned 1,082,265 bachelor's degrees in comparison to men earning 812,669. Degree earning differences are seen at graduate levels of study as well with women earning 452,118 master's degrees, and men earning 306,590 master's degrees, and in the awarding of doctor's degrees women were represented with earning 93,626 doctor's degrees and men with 84,921 doctor's degrees. Data such as this presents a clear picture that women are now represented at beyond equivalent numbers in the academy as students, so why are women not represented in equal numbers in higher education leadership? The study presented in this article seeks to examine this phenomenon as it relates to the dearth of women trustee board chairpersons at the helm of private colleges and universities. To answer the question of why this discrepancy persists is difficult; however, the answer is worth seeking. A starting point in the examination of this dilemma is determining the leadership philosophies and roles, in particular that of the board of trustee chair, to further illuminate the issue. Literature and research in the areas of women, leadership, and governance is plentiful; however, research in the area of governing board composition and gender is inadequate. To that end, this work presents a snapshot of findings from a study that examines some of the issues regarding women and leadership, and their pathways to the position of board chair at independent institutions.

Former Secretary of Education, Margaret Spellings' Commission on the Future of Higher Education placed heavy emphasis on higher education accessibility. The Commission cited diversity as a non-academic barrier to accessibility. This provides support for the importance of diversity on college campuses. Students who see diversity represented at higher levels of administration may be encouraged to pursue opportunities for furthering their education; alternately, students who do not see individuals like themselves reflected in the governing boards of our nation's institutions may feel yet another barrier to their inclusion in the academy (United States Department of Education 2006).

The 2010 Association of Governing Boards of Universities and Colleges (AGB) survey on board composition reflected the inequity in the number of women represented in the board of trustee chair role. The board composition survey conducted by AGB showed that women comprise less than one third of governing board membership at independent and public institutions of higher learning. The study also showed a slight decline in representation of women on public governing boards from the previous two board composition surveys conducted by AGB with women making up 28.4 percent of the membership compared to a previous percentage of 29. In contrast, there was a slight increase on independent governing boards with women being represented at 30.2 percent, up from 28.4 percent in 2004. While this data shows an increase in women serving as trustees at independent institutions, men continue to hold a greater share of board chair positions at independent institutions at 81 percent, and hold 82.6 percent of board chair positions at public institutions (Schwartz 2010). These numbers reflect a history of inequity regarding women's attainment of higher level leadership roles in academe.

Based on the aforementioned issues, the purpose of this study was to determine the career paths, obstacles, and facilitators of women trustee board chairpersons in order to develop a paradigm of ascension patterns. Exploring the executive level experiences of women in the role of board chair will add to the body of knowledge regarding leadership and gender at the policy-making level. The plethora of definitions regarding leadership and leadership styles contributes to the challenges of investigating this issue. Researchers have sought to identify issues regarding gender and leadership from various perspectives, ranging in viewpoints from feminist ideology to business models, and studies, both qualitative and quantitative in nature have sought answers to the questions surrounding women and leadership roles.

Research and litigation related to parity and equality demand that as the population of participants in higher education becomes more diverse, leadership should reflect this diversity range as well (Cottrol et al. 2003). Historical precedent related to the powerful role of external influences on student

success was established when Kenneth Clark's psychological research became critical to the landmark Brown decision (Brown v. Board of Education, 1954). It is important that the leadership at colleges and universities reflect the students that are being served. While educational leaders instill in students the belief that education can serve as a gateway to opportunity, students may doubt the veracity of the assertion if they do not see themselves reflected in the school's cadre of leadership (Hurtado et al. 1999).

As the previous generation of higher education leaders make preparations for retirement, it will become increasingly important to maintain a healthy pool of quality leaders from all backgrounds. As the "old guard" of academe leave their posts, the importance of having qualified and competent leaders to assume the roles left vacant will be significant. Additionally, 15.6% of institutions report a mandatory age retirement policy, with an average age of 72 (Schwartz and Akins 2004). With the increasing age of trustees, many institution's governing boards may soon find themselves in positions to rebuild their boards with new members. With the increase in women as students and graduates, the alumni pool from which trustees are selected will grow increasingly diverse. It is essential that leadership opportunities are in existence for one of the largest segments of the population. A diverse pool of leaders can lead to diversity amongst ideas regarding leadership. This diversity may be capable of creating insight into how to best approach the issues of the future that will impact higher education.

The findings of this study are significant to those concerned about gender inequality in higher education administration leadership. Additionally, findings from studies such as this one can provide insight into leadership practices in higher education and assist with the development of governing board leadership programs as it pertains to governing board composition, training, and selection process.

The findings presented in this article stem from a study set in a southeastern State in the United States. Two main factors contributed to the selection of this State. In recent history, southeastern States have had the largest gender gap in college enrollment, with an average enrollment ratio of 1.4 males to 1.0 females (United States Department of Education 2006). Additionally, this southeastern State was identified by the author as having the second largest population of female board chairs at independent institutions in the Southern Association of Colleges and Schools (SACS) eleven state accrediting region.

The study was qualitative in nature and utilized a phenomenological analysis. This methodology describes the meaning of the lived experiences for several individuals about a concept or phenomenon (Creswell 2007). The researcher identified independent institutions in a southeastern State with a female governing board chair. Individuals from these institutions were selected for participation in the study. The researcher identified the institutions in the southeastern State with past and present female board chairs though a search of the web sites of each institution. The researcher then contacted the institutions via telephone to confirm the identity of the board chair. Each female chairperson was contacted via email to gauge their interest and availability to participate in the study. The researcher confirmed participation from the respondents via telephone and mailed letters of intent and details regarding the study to the participants. The participants and researcher then confirmed dates, times, and locations for the in depth interviews to take place. An initial interview was conducted with each participant. An interview guide was used to assist the interview process. The interviews were conversational in nature, allowing for the experience of the participant to emerge. Notes regarding participant non-verbal responses were taken during the audio recorded interviews. The interviews were transcribed following each interview session. The researcher sent the resulting transcripts to the participants for review, and follow-up interviews were conducted when necessary to confirm and clarify responses. The researcher coded the transcript to identify themes which were used to address the four research questions.

The study was guided by the following research questions that were designed to gather information based on the "voices" of women who served in the positon of chair.

1. What themes will emerge when the female governing board chairs describe their experiences?
2. Will the female board chairs identify any barriers or obstacles to their success? If so, what are they?

3.  What themes will emerge in relation to overcoming any identified barriers/obstacles?
4.  What factors are identified as contributors to the attainment of the board chair role?

These questions represent core inquiries that will evolve into additional questions that emanate naturally from the "voices" of the target population. Some of those interview questions included the following: Is there a particular experience that you would like to share with me that you would describe as pivotal in your journey to board chair? Can you share with me personal experiences or individuals who have influenced your attainment of the board chair role? Are there key individuals that contributed to your successful attainment of the board chair role? Are there particular events or experiences that you would identify as contributors to your successful attainment of the board chair role?

The author utilized a phenomenological approach to explore the participant's ascension patterns and experiences in relation to achieving the board chair position. To delve into the experiences of female board chairs, the author conducted in depth face-to-face (when possible) or telephone interviews, transcribed the interviews, and employed the code-recode method to ensure reliability. Follow up interviews for additional clarification were used when necessary. These methods allowed the author to ascertain the richness of each participant's experiences. The subsequent findings provided within this article provide insight into leadership practices in higher education and are intended to assist with the development of governing board leadership programs as they pertain to governing board composition, training, and selection processes.

## 2. Results

### 2.1. Biographical Sketches of the Board Chairs

To understand the development of the skills that have equipped each respective Chair for her current successes, it is important to have an awareness of the characteristics of these individuals. All the Board Chairs at the centerpiece of this article served as female Board of Trustee Chairpersons at independent colleges and universities in a southeastern State.

The collective of Chairs includes five women. The women range in age from their early 60's to early 70's.

Chairperson One is an alumna of the institution where she serves as Board Chair. She is the third alumna to serve in the role of Chairperson in the history of the college. She has had an extensive career as a human resources executive, at the time of the interview her professional role was that of the president and principal consultant of her own human resources consulting firm. She has strong familial ties to the college and set a childhood goal at the age of five to attend the institution she now serves as Chairperson.

Chairperson Two does not have alumni or familial ties to the institution for which she serves as chairperson. She is a self-described career volunteer, and in her words has not worked "in a career to speak of". She has however, served in a variety of high level volunteer positions in the southern area of the southeastern State. She has a passion for community involvement that led her to her years of service at College Two.

Chairperson Three has a familial connection to the institution where she served as Chairperson. The institution is her mother's alma mater. Chairperson Three is an attorney by trade; her current professional role is that of co-owner along with her husband, of a prominent luxury car dealership in the southeastern State. She has an extensive background serving in a variety of leadership roles in State k-12 education and higher education.

Chairperson Four is an alumna of the institution where she served as Board Chair. She has held a variety of leadership roles at her alma mater, ranging from volunteer alumnae roles to high-level administrative roles. Chairperson Four is currently retired. She demonstrates a high level of commitment to her alma mater through a long and varied relationship with the institution.

The fifth and final chairperson is an alumna of the institution for which she served as board chair. She is an attorney by trade, and has been employed as a law school administrator for the majority of her professional career. Chairperson Five has had a longstanding relationship with the college as a Trustee and Board Chair.

These descriptive character vignettes offer a glimpse into some of the components that inspired each woman's path to the role of board chair. The impact of their respective backgrounds and the context and era in which they came of age become evident as they describe the skills that they feel have been pivotal in reaching their position as board chairpersons. These skills, which often contrast with conceived notions and stereotypes of women leaders, equipped these women with the ability to successfully obtain board of trustee chair appointments. Nuanced inquiry into these factors allows for a more informed picture of the success and hindrances to success that women face in their pursuit of executive levels of leadership.

*2.2. Discussion of Findings*

In analyzing the stories of these five women, two themes were identified in relation to factors identified as contributors to the attainment of the board chair role: (a) possessing skills needed to successfully fulfill the chairperson role and the use of those skills/talents as prior board members to bring about positive change to their respective boards; (b) having been identified as leaders by key individuals at their respective institutions, both when they were being considered for the board and when they were being considered for board chairperson. These themes provided the author with insight into the ascension patterns and experiences of past and present female board of trustee chairpersons at independent colleges and universities. While two overall themes were identified as contributors to the attainment of the board chair role, this article will focus on the skills that each identified as being instrumental in their ascent to the board chair role.

While there are numerous areas for exploration regarding the leadership journeys of these women, this article will focus upon the first theme, the self-identified skills that contributed to these women being named to the role of board chair. The focus on these skills may be helpful to those women who are seeking similar leadership roles. By thoroughly examining and exploring this subset of skills individuals who desire board chair roles are provided with a potential list of skill sets to develop that may assist in attaining their leadership goals.

2.2.1. Possessing Skills Needed to Successfully Fulfill the Chairperson Role

The chairpersons indicated skills that they possess that were key contributors to their respective ascensions to the chair role. Some of the women identified these skills as "skill sets". The women agreed that these skills were integral to being selected to serve as chairs and their subsequent success in the chair role as well as other leadership capacities. Additionally, the women referenced examples of how they used these skills to bring improvement to their respective boards in the capacity of board members prior to their service as chairs.

2.2.2. Skills Needed to Serve as Board Chair

Each woman came to the chairperson role in their own way, however there was a shared theme in that each individual identified specific skills they embodied and that subsequently equipped them for leadership as a board chair. The Chairpersons indicated that they worked at developing their skill sets. Chairperson One participated in leadership programs developed for trustees that were helpful to her gaining the skill set needed to serve in the capacity of trustee and chair, and she shared the following,

> I think the thing that I will say is, in preparing myself, I got involved with AGB (Association
> of Governing Boards). It's an organization of governing boards, so I am now on their board.
> So, I got involved with them. In fact, not only am I on their board, but they have me
> teaching in some of their programs. This intentional effort to develop this skill set was seen

as an opportunity to equip one's self with tools to strengthen the capacity to lead the board. Each chair agreed that there were leadership skills that enhanced their capability to serve as chair and to improve existing practices of their respective boards.

In speaking with Chairperson Two, her experience reiterated the fact that trustee boards seek individuals with particular leadership skills. Chairperson Two not only identified her own skills, but stated that when College Two searches for new trustees they look for individuals who possess certain skill sets as well,

> So, you can bring a love and a passion and motivated spirit to things, but you will never have the respectable skill until you work through the nominating process or the Board development process. That's one of the things we do really well now at College Two; we are intentionally and specifically targeting individuals from certain locations and with a certain skills set, so that you have a talented group of people to hold the school in trust.

She identifies among those skills, her ability to rally individuals together to utilize their passion and energy to bring about positive change in the community and civic organizations. This notion of boards intentionally seeking individuals with particular skill sets provides evidence of the importance of developing leadership skills amongst women to serve as board chairs and to do so with intention.

Many of the Chairs indicated that there were transferable skills that contributed to their success as Chair-persons and in their professional roles. Chairperson Three identifies some of the skill sets that she has refined throughout the years that have been useful to her in business and in her civic leadership roles,

> One basic skill is simply how to hold a meeting; how to set up a meeting agenda that's going to be conducive to your Board and productive. Trustees around the country don't want to spend 24/7 in a Board Meeting. So, getting to a point where you really know how to set that agenda and get other people motivated is crucial. This last meeting, our board liaison told me we had nearly 100 percent attendance, which is more than we've ever had with a large board of 30 people. So, at those meetings, you need to make sure something is happening at that Board Meeting.

One of the first things that I first did is to feature a professor or a student at every meeting. At this past meeting I featured a professor and a student, and the Board was so impressed and enthusiastic. They go back to their positions so pumped, and they know what they're working for; you work for this student.

Look how bright and sharp he/she is, and therefore we need to support this endeavor. So, that whole meeting management is one of my particular fields and how to keep people pumped and passionate about the university. Make sure you've carved out your skill of what you do well for the institution. As this Chairperson indicated she developed a skill that she already possessed. Additionally, she made herself known for that particular skill and demonstrated to the institution and the board how this skill was beneficial to the progress and success of the board.

This emphasis on bringing improvement to board functioning through implementing particular practices of leadership continued in the account of Chairperson Five's experience. Chairperson Five references how her skill set brought improvement to the board, noting,

> If you talked to the people who were on the board when I started as chair, I think they would tell you the most noticeable differences were the meetings ended on time, that they got their materials more in advance and they had better information, we moved to a consent agenda so we did not keep doing the same things over and over and we tried to move to a place where people's time was well spent and were intellectually invested when they were there. These identified skills assist in creating a catalog of skills that other individuals seeking the chair role can develop in their quest towards leading boards.

## 2.3. Networking and Networks

Wolff and Moser (2010) indicate that networking has been shown to be positively related to promotion. They further state that networking also provides an outlet for information sharing which can assist with establishing alliances and becoming more visible in an organization. The relationships formed through acts of networking may in some instances provide opportunities for added access. The experiences of the women in this study supported this perspective.

While several of the women recounted their stories of "accidental ascension" their accounts often circled back to professional relationships and opportunities to develop connections with their respective institutions through networking and networks. Chairperson One, an alumna of the college where she serves as Chair, spoke of her membership in a number of community organizations prior to joining the Board of Trustees. She indicated that she became known as a leader amongst her network in the community and when opportunities such as the one to serve as a trustee arose, Chairperson One was often at the top of the list of candidates. Chairperson Four, also an alumna Chair, became known in her networks and interactions with others as an individual who could come into an organization to right wrongs and turn things around. While the women described their respective ascensions to the role of chair as unintentional, it is evident that their reputations for successful leadership amongst their networks were instrumental in their rise to the chair position. This ability to network provides opportunities for women to develop and express their agency in organizations. By developing a network of women who are skilled leaders a number of women benefit. These networks enable the opportunity for adding women to the legions of leaders through providing a "pool" of leadership candidates. This is critical in succession planning and in advocating to bring more women to the table.

## 3. Skill Development

### 3.1. Familial Influence on the Development of Leadership Skills

The Chairpersons recognized that the development of their respective skill sets began in girl hood and they recount with ease the impact of their own families in their attainment of leadership roles. In the development of self-identified skill sets, the women noted the significant impact of their upbringing. One of the key themes that was identified in regards to contributors to the attainment of the board chair role was that of the influence of family and influential family members were referenced in many instances. In regards to the socio-economic background of their childhood family, many of the Chairpersons indicated that due to the "way" they grew up, success and education were expected. Expectations for success were established early and reinforced often by a variety of family member. This expectation for success and achievement served as a life-long influencers to affect change through leadership. Chairperson One was particularly influenced by an aunt, who not only inspired her to attend college, but became an inspiring force in her becoming a trustee and ultimately chair of the board,

> So, I was born in a family of three girls. I have an identical twin sister, born and raised in Los Angeles and my mother was a school teacher in Watts. We lived in Watts in Los Angeles, and my father was the local criminal lawyer in the area. I always knew that I was going to college. I come from . . . on my mother's side in particular, were well educated people. So, I always knew that I was going to college, and I always knew that I would be successful, and so, I was in a family that was focused on education as most middle class black families were in those days.

Chairperson One had this to say about her aunt who influenced her to give back to her college by serving as a trustee and chair,

> And so, what got me to College One, was I had an aunt, and this particular aunt had gone to College One, and I just knew because of her, when I was four or five years old that I wanted to go to college. I remember just being blown away by her as a person and she

became, obviously, an impactful person in my life. And, when I was four or five years old, she took the train from Atlanta to Los Angeles, and I remember her getting off the train and thinking I want to be just like her.

When asked the question of who influenced her to become involved with College One as a trustee, Chairperson One replied with the following:

So clearly my aunt was very influential on that. She was very involved, and she could see the leadership that I was doing, with these corporations that I was working for and she's saying, "You've arrived at all these big-time jobs, and why can't you give College One some of that knowledge?" So, she helped me to think about that. This encouragement from a trusted family source provoked Chairperson One to see herself in a new light, as an individual who could have an impact and make a difference.

Chairperson Two recalls her childhood family situation as being influential to her development as a leader early on,

I have found myself in leadership positions in a number of different venues. And, as I ponder and go back . . . as the older of two children, my father died when I was 13, and my mother was not a real strong person emotionally, and so, I became a decision maker sort of early in life. In high school, I laughingly say, that even in groups of teenage girls they would say, "Now what are you going to say Mother Smith" (my maiden name) and they would look to me to keep them from getting into too much trouble and smoking too many cigarettes behind the barn door.

The lived childhood experiences of Chairperson Two illustrate how these experiences were impactful in her leadership development. As she recounts stepping in to serve as a leader in her family upon her father's death, later in her life she displays the same skill of stepping in and assuming leadership when various organizations that she is involved with have a need for leadership to arise.

Chairperson Three attributes her involvement with college three directly to her mother, a graduate of the institution. She recalls a similar upbringing to Chairperson One, growing up with an educator mother and attorney father, "My father was a lawyer, and my mother was a school teacher, an educator who went to College Three here in this southeastern state." Referencing her grandparents,

I often times think that these seemingly different environments with my grandfather being a physician, my father being a lawyer, but with the farm experience (maternal grandparents) where there was no running water, wood stove and seemingly opposite experiences really converged on me and my other siblings. So, the business leanings came from both sides of the family, because my grandfather was a physician as well as a pharmacist before, so he ran a business. He ran a pharmacy and also his medical practice. Those two seemingly divergent paths really converged as I grew up, and I really believe in earnest served as a foundation for who I was to become, a business owner and operator, an entrepreneur, a community activist (if you will), and for love of my mother's institution, from which she graduated in 1938. So, all that background, sort of, came and had an impact on my life as to who I became.

She continues the conversation by sharing her reason for the strong connection to College Three, she had the following to say regarding her commitment to College Three, "I am passionate about it, my mother went there, and I saw that that's the foundation for who I am today."

Chairperson Three's experiences echoes the sentiment shared by her fellow chairpersons. It is the love and encouragement from family that inspired the same love and commitment to servant leadership at their respective institutions. The interactions with and encouragement from family proved to be a vital factor in the development of each individual's leadership skills.

### 3.2. Establishing One's Self as A Leader

The study identified that the chairpersons did not intentionally seek the chairperson role, in most instances they were asked by key individuals (such as the president or former board chairs) at the institutions that they serve. The chairpersons were able to identify individuals who had identified their talent to serve, be it at their institution or elsewhere. This identification of talent was integral to the trustee being appointed to their chair role.

Chairperson One identified her fellow board members as being pivotal in noticing her skill as a leader,

> It became obvious to board members that I was a natural leader. I asked all the right questions, I put in the time, and I did my homework. I worked at the things that needed to be worked at; in this case I was Chair of the Board of Affairs Committee. And then I chaired the Search Committee for the President.

Chairperson Two has the longest standing relationship with the institution that she serves as chair. She indicates that she was identified as a leader who's service would be beneficial to the college through her numerous volunteer efforts with other organizations. She speaks of her many years of service to College Two, "As for the college, I was asked at some time nearly thirty years ago to join the Board, and have stayed on the Board. I have been the Chair now for about ten years."

Chairperson Three reveals in her interview that she was asked by two individuals to serve at college three in different capacities, once by a former fellow board of regents trustee to serve as a College Three trustee and by another colleague to serve on an advisory board at the institution. Chairperson Three shared the following about her being identified to serve as a leader at College Three,

> So, I started out serving on that Advisory Board Council, which is something that may not be as typical that you start out on the advisory council for a particular school instead of communications or whatever, but I did start out on the Business Advisory Council. The school president said that this Business Advisory Council was stronger than the Board, and wanted to get us on his Board. And he asked me to serve on the Board, and I did . . .

After many years as an alumnae volunteer and a past employee of College Four, Chairperson Four was identified by the college president as a leader to serve the college in the capacity of trustee and ultimately she became the chairperson, "And in 2003, the president of the college asked me to join the Board, and I did that and enjoyed it very much. I became Board Chair in May of '08."

In addition to being identified to serve the institution initially as a trustee, Chairperson Four was also recognized by her peers when the time to appoint a chair arrived,

> The board elects the chair. The Board had selected me to serve, but the President certainly influenced that and asked me to take the position, but there is a committee of trustees that nominate the officers of the Board. That committee includes the president and the current board chair . . .

Chairperson Five indicates that she was identified to serve initially as a trustee by a chair, and ultimately as a chair by an outgoing chair and her fellow trustees. Chairperson Five shares her experience of being asked to serve as a trustee through connecting with the board chair at a cocktail party,

> In parallel to that . . . my husband is in a big law firm, senior partner in which is the chair of the board at College Five, and so he and I are at a cocktail party and I say to . . . you know I am really irritated at the college because they have refused to give tenure to a Jewish woman, and he said that is not quite right, and I said well what is quite right and so he starts explaining to me kind of what's going on and the inability to have a community that can span beyond this very limited concept, Christian entitlement just really irritates me

> and the next thing I know he says to me, this is probably a year later he says to me how would you like to be on the board to see if you can make it better and I say ok . . . because I can do anything . . .

These examples of recognized leadership offer additional reinforcement of the importance of not only developing one's leadership skills but the opportunity to practice those leadership skills. In having the capability to make an impact and a difference on their individual boards these chairpersons further establish themselves as leaders in the field.

## 4. Future Implications

### 4.1. Impact of Skills on the Advancement of Women as Chairs

As illustrated by the diverse collective of women who participated in this study the impact of leadership skills that are possessed and utilized while guiding a board are critical to the advancement of women as chairs. It is this identified array of leadership skills that have proven pivotal in these chairpersons ascending to the role of chair from the group of their peers. Their ability to develop, capitalize upon and utilize their respective skills sets them apart from the crowd. While the chairs may not have felt that their pursuit of leadership was intentional it is clearly illustrated through their experiences that they exhibited their leadership prowess in such a way that ascension to the chair role was inevitable.

### 4.2. Using Your Skills to Influence Positive Change

Governing Boards have a multi-faceted role and purpose. The joint statement on academic institution governing boards states the following about the role of governing boards (Duryea and Williams 2000):

> Governing boards in higher education institutions operate as the final institutional authority. Governing boards have a special obligation to assure that the history of the institution serves as a prelude and inspiration to the future. Additionally, the board helps to relate the institution to its primary stakeholders. The governing board of an institution, while maintaining a general overview, entrusts the conduct of administration to the administrative officers, the president and the deans, and the conduct of teaching and research to the faculty. This statement reiterates the importance of the board and the weight of responsibility that comes with board leadership. As indicated by the chairpersons it was one thing to possess strong leadership skills but another thing to utilize those skills to affect positive change.

Information garnered from this study provides insight for women aspiring to the role of board chair as well as boards that seek to appoint more women as chairs to their institutions' trustee boards. The respective board chairs resoundingly supported an institution's attention to appropriate training for those who were being considered for the role of chair. This training ranged from one on one mentoring and shadowing with an institution's president to time spent at retreats and national training meetings focused on the role of the trustee and collegiate governance. The chairpersons collectively offered the advice that women seeking the leadership role of chair focus on developing skill sets centered around fiscal knowledge, team building, and organization. In all instances the chairpersons agreed that institutions must be intentional about succession planning and identifying and preparing individuals to serve in the role of chair. This intentionality coupled with passionate and prepared individuals provide the key for successful boards and leaders.

## 5. Conclusions

(1) Due to the critical role that boards play in establishing policy and maintaining fiscal accountability, it is essential that the trustees and chairs possesses certain skills. The chairpersons in this study

each referenced possessing certain skills or skill sets that assisted them in their ascension to the chair role. These skills ranged from the ability to set appropriate meeting agendas to having the business skills and knowledge to turn around a financially struggling institution. One chairperson referenced how her skill to effectively plan meetings had substantially increased the board meeting attendance and engagement at the meetings. Another chair indicated her knowledge of diversity issues and her ability to bring individuals together in spite of their differences. Her skill and passion in this area led to the development of a non-discrimination policy for the college.

(2)　Possessing certain skills was deemed critical by all of the chairpersons for a successful ascension to the chair role. As each of the chairs referenced the skills that they brought to their role, they also emphasized that possessing these skills and skill sets was critical in their succession planning. As chairs depart from their roles and seek new trustees each of the chairs indicated that they look for individuals who possess these skills when inviting new trustees.

(3)　All of the female chairs share the common threads of being held in high esteem by their peers, and having served in a variety of leadership positions during their lifetimes. These characteristics have equipped them to assume the mantle of leadership at the executive level. All of the women have ingrained in them the tenants of servant leadership, serving not only the institutions that made them who they are today, but also serving institutions that they will never directly benefit from.

(4)　While women continue to be underrepresented in the role of chairperson in comparison to their male counterparts, these women have successfully attained the role of chairperson. The experiences of these women vary, yet commonalities can be found amongst their experiences. These experiences provide insight into the journey of the women to the chair role and present a paradigm of ascension patterns for others to follow.

**Conflicts of Interest:** The author declares no conflict of interest.

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
