# Peer review of "Ascending: An Exploration of Women’s Leadership Advancement in the Role of Board of Trustee Chair"

_admsci, doi:10.3390/admsci8010007_

Round 1

Reviewer 1 Report

MS # admsci-259495

Ascending: An Exploration of women’s leadership advancement in the role of Board of Trustee Chair.

I found this paper very interesting in that it depicted intriguing stories of women in successful leading roles with their own voices. This is a timely study that helps shed light on women leaders’ ascending paths.

With that said I do have a number of concerns as detailed below and hope you’d find them helpful.

1.     The framing of problem of inquiry and research question(s): The introduction section was very descriptive and detailed but the statements on why this study mattered and what research questions you set off to inquire were not explicit and rather buried. I think more space should be allocated to these two components in this section – what’s the background that inspires your inquiry. The selection of this particular research design can only be justified when the purpose of study is sufficiently presented.

2.     Research design: why this particular design was chosen, what advantage it offered, and summary of sampling and data collection procedures should be presented however briefly. What types of interviews were conducted? What (types of) questions did you use in interviews?

3.     Data presentation and discussion of findings: The study identified one theme contributing to chairwomen’s leadership path: possessing necessary (self-identified) skills.  The quotations from five Chairpersons actually demonstrated a number of diverse (sub)themes, some of which captured by the discussion lines (Line  168, 174, 187). These subthemes should be more logically organized under one overarching theme - if “possessing necessary skill” was that ONE central theme as you stated. All subthemes should be presented in a more coherent way. For instance, under the theme of “possessing necessary skills,” Subtheme 1: answer the question of “what are the necessary skills?” (e.g., conference, networking); Subtheme 2: answer the question of “how are these skills developed?” (e.g., familial influence); Subtheme 3: answer the question of “how have these skills brought advancement” etc.

4.     Discussion should be more in-depth and focus on fleshing out how each story and quote demonstrated the themes/subthemes and what logical relationships were among them. Current paper is overwhelmed with quotes while authors’ analysis and insights on what we can learn from these stories have been minimized to section titles. As a result, the contribution of the study was also minimized.

5.     Pg 4, Line 124 “this article will focus upon the first theme, the self-identified skills that contributed to these women being named to the role of board chair” – yet Pg 7, Line 248-249 “Chairperson One identified her fellow board members as being pivotal in noticing her skill as a leader” – isn’t this the second theme that this article was not going to cover?

6.     Conclusions: these section (pg 8 – 9) should be reorganized to stay align with research questions and (sub)themes identified and how the findings are related to the bigger literature. 

Author Response

Dear Reviewers,

Please find a memo attached which outlines how I have addressed the requested revisions. 

Respectfully Submitted

Reviewer 2 Report

Beef up the methodology section a little bit.

The study is based on narratives of five women leaders and it would be interesting to know a little bit more about them, i.e background details that will help the reader understand what is unique about these experiences.

Provide some justification for the methodological choices

It doe not tell the reader much to just say "follow up interviews for additional clarification were used where necessary". Provide some details

Theorize the discussion a little more by bringing the literature. The findings are interesting but are just at description level.

the contextual background information referred above could also be brought into the discussion to illuminate

Author Response

(The authors gave the same response as above.)

Round 2

Reviewer 1 Report

There is a significant improvement in this revision. The issues and questions raised have been addressed sufficiently. Appreciate your effort. 

Reviewer 2 Report

The comments have been satisfactorily addressed.